# Structure of Fe-Mn-Al-C Steels after Gleeble Simulations and Hot-Rolling

**DOI:** 10.3390/ma13030739

**Published:** 2020-02-06

**Authors:** Liwia Sozańska-Jędrasik, Janusz Mazurkiewicz, Krzysztof Matus, Wojciech Borek

**Affiliations:** Silesian University of Technology, Department of Engineering Materials and Biomaterials, 18a Konarskiego Street, Gliwice 44-100, Poland; Liwia.Sozanska-Jedrasik@polsl.pl (L.S.-J.); janusz.mazurkiewicz@polsl.pl (J.M.); Krzysztof.Matus@polsl.pl (K.M.)

**Keywords:** Fe-Mn-Al-C steels, thermomechanical processing, microstructure, κ-carbides, M_7_C_3_, (Ti,Nb)C, EBSD, gleeble simulations, hot-rolling

## Abstract

In this paper, analytical results are compared for the newly developed steels, Fe-Mn-Al-C (X105) and Fe-Mn-Al-Nb-Ti-C (X98), after being hot-rolled and also after undergoing thermomechanical treatment in a Gleeble simulator. These steels have a relatively low density (~6.68 g/cm^3^) and a content of approx. 11% aluminum. The multistage compression of axisymmetric samples constituting a simulation of the real technological process and hot-rolling performed on a semi-industrial line were carried out using three cooling variants: in water, in air, and after isothermal heating and cooling in water. The temperature at the end of the thermomechanical treatment for all variants was 850 °C. On the basis of detailed structural studies, it was found that the main mechanism for removing the effects of the strain hardening that occurred during the four-stage compression involved the dynamic recrystallization occurring in the first and second stages, the hot formability and dynamic recovery in successive stages of deformation, and the static and/or metadynamic recrystallization that occurred at intervals between individual deformations, as well as after the last deformation during isothermal heating. Analysis of the phase composition and structure allowed us to conclude that the tested steels have an austenitic-ferritic structure with carbide precipitates. Research using scanning and transmission electron microscopy identified κ-(Fe, Mn)_3_AlC and M_7_C3 carbides in both the analyzed steels. In addition, complex carbides based on Nb and Ti were identified in X98 steel; (Ti, Nb)C carbides occurred in the entire volume of the material. Slow cooling after thermomechanical treatment influenced the formation of larger κ-carbides at the border of the austenite and ferrite grains than in the case of rapid cooling. The size and morphology of the carbides found in the examined steels was varied. Back-scattered electron diffraction studies showed that wide-angle boundaries dominated in these steels.

## 1. Introduction

Due to their excellent mechanical properties, relatively high plasticity, and forecasted relatively low production costs (e.g., no heat treatment), Fe-Mn-Al-C steels, which are the subject of this analysis, can be potentially used for elements of transport infrastructure and vehicles such as cars, buses, or trains, as well as in all kinds of constructions where the weight of the structure is one of the most important criteria for the selection of materials [1,2,3,4,5,6,7,8,9,10,11,12,13,14,15,16]. In recent years, there has been a development of steels with reduced density, which include Fe-Mn-Al-C steels with a 15% lower density compared to typical structural steels [4,17,18,19,20,21,22,23,24]. In their papers, Chen et al. presented the division into different structures of Fe-Mn-Al-C steel with reduced density after hot-rolling (Figure 1) [1,2,4]. They described that, depending on the content of alloying elements such as Al, Mn, and C, the structure of these steels can be ferritic, ferritic-austenitic (ferrite-based duplex steels), austenitic-ferritic (austenite-based duplex steels), austenitic-ferritic with κ-carbides, or austenitic (Figure 1) [1,2,4]. 

Rana et al. described multiphase TRIPLEX Fe-Mn-Al-C steels with a structure consisting of three phases: austenite, ferrite, and κ-(Fe, Mn)_3_AlC carbides and others. They found that κ-carbides are formed from areas enriched in carbon through spinodal decomposition and are key determinants of the properties of these steels [25,26]. The precipitation of Fe_3_AlC_x_ carbides in Fe-Al-Mn-C steels can occur in both austenite and ferrite, depending on the content of the alloying elements. In Fe-Al-Mn-C alloys with a high manganese content (above 10%), Mn atoms take the place of Fe atoms in the above carbide, and the stoichiometry of the Fe_3_AlC_x_ carbide is transformed into (Fe, Mn)_3_AlC_x_ [18,25,26]. Etienne et al. analyzed transformations in Fe-Mn-Al-C steels based on differential thermal analysis to understand phase formation mechanisms. Figure 2 shows a scheme for continuous cooling of steel: Initially austenite is formed from ferrite, and then κ-carbides form at the boundaries of austenite and ferrite. During further cooling, austenite is transformed into ferrite, and κ-carbides are released as a result of eutectoid transformation [25,26,27].

Chen et al. showed that, in Fe-Mn-Al-C steels, recrystallization occurred during a traditional hot-rolling process, and the microstructure after hot-rolling showed elongated austenite grains with annealing twins [1,28]. After slow cooling of the steel (air-cooled), κ-(Fe, Mn)_3_AlC carbides and α-ferrite were found along the austenite grain boundaries and inside the austenite. The κ-carbides were then three to six times larger than in the case of fast-cooled (water) steels, while reducing the plasticity and strength of these steels. Therefore, with the current state of knowledge about the structural mechanisms of this steel group, it is recommended to quickly cool Fe-Mn-Al-C steel to avoid the formation of large κ-carbides [1,28,29,30]. In Fe-Mn-Al-C steels, two types of κ precipitations are observed: Intragranular (fine, increases the yield strength) and intergranular (larger κ-carbides, which can lead to a significant deterioration in steel yield) [1,2,3,6,7,8,10]. 

In describing the structure of Fe-Mn-Al-C steel after hot-rolling, Bausch, Frommeyer et al. noted the formation of ferrite bands [2,4,31]. The high-aluminum content tended to lead to the formation of ferrite bands parallel to the rolling strip surface in the direction of hot-rolling. The ferrite bands were concentrated in the center of the rolling mill, where the concentration of austenite-stabilizing elements (manganese and carbon) decreased by impoverishing the crystallization front with austenite-forming elements during crystallization [2,4]. Literature reports confirmed the recrystallized austenite microstructure with thin ferrite bands in the analyzed steels, and also identified AlN (aluminum nitride) precipitates and carbides at grain boundaries [2,4].

The mechanical properties of high-manganese steels can be shaped by suitably-selected thermomechanical treatment, which is the preferred method of producing mass products in economic terms. The use of appropriate heat and plastic deformation treatment allows the best relationship between the strength properties and plasticity of these steels to be obtained [11,20,21,22,23,24,32,33,34,35,36,37,38,39]. The introduction of thermomechanical treatment methods enables the integration of technological production lines of metal materials, including the preparation of charge materials, the smelting and purification of liquid metal, together with the continuous casting of ingots and their hot plastic processing with controlled cooling from the temperature at the end of this treatment [11,12,13,14,15,16,17,18,20,21,22,23,24].

## 2. Materials and Methods 

The tests were performed on experimental high-manganese X98MnAlNbTi24-11 (X98) and X105MnAlSi24-11 (X105) TRIPLEX steels, whose chemical composition is given in Table 1. These steels are characterized by a high-metallurgical purity, associated with low concentrations of S and P impurities. The melts were modified with three rare earths: cerium, lanthanum, and neodymium. The chemical composition of the tested steels was selected to obtain a multiphase structure based on austenite and ferrite with carbides. A controlled concentration of Nb and Ti micro-additions with strong chemical affinities for nitrogen and carbon was introduced into the X98 steel. These additions were designed to limit the growth of austenite grain and increase the strength due to the precipitation processes of carbides of these elements.

The tested steels were cast in an argon atmosphere into a cast iron ingot, and then preliminary hot working of the ingots was carried out using the free forging method on a Kawazoe high-speed hydraulic press with a pressure of 300 tons. The forging temperature was from 1200 to 900 °C with inter-operative reheating. Samples for structural tests were cut out of the forging material in the form of flat bars 32 mm thick and 200 mm wide, and 10 × 12 mm cylindrical samples were prepared for thermomechanical treatment tests (Figure 3a). These tests were performed using a Gleeble 3800 thermomechanical simulator from DSI (Dynamic System Inc., Poestenkill, NY, USA), equipped with a direct resistance heating system, which maintained the set temperature with an accuracy of 1 °C. Thermocouples read the sample temperature, and the system reacted to temperature changes using resistance heating. The Gleeble 3800 system was equipped with a hydraulic mechanical system allowing for the application of pressures of the order of 200 kN during compression, enabling tests to be carried out with a deformation rate in the range of 0.0001 to 200 s^−1^. Linear variable displacement transducers and strain gauges for measuring pressure allowed feedback to be obtained, which enabled accurate performance and high repeatability of the set mechanical quantities of the planned process. To reduce friction, graphite and tantalum foils were used between the sample surface and the anvil surfaces, while both surfaces were coated with high-temperature nickel grease (Figure 3b).

The samples were resistively heated in an argon atmosphere at a rate of 3 °C/s to a plastic deformation temperature, T_d_, of 850, 950, or 1050 °C and kept at that level for 30 s to equalize the temperature throughout the entire sample (Figure 4). The plastic deformation rates for the compressed samples were $\dot{\varepsilon }$ = 0.1, 1, and 10 s^−1^, respectively.

A four-stage hot compression process of the abovementioned samples was designed and performed on the Gleeble 3800 simulator (Figure 3a), simulating the final rolling process. Degree of deformation, plastic deformation rates, and break times between successive plastic deformations (Figure 5) were selected taking into account the conditions of the planned real semi-industrial hot-rolling of flat bars with an initial thickness of about 4 to 5 mm for 2 mm thick bars. In addition to determining the force-energetic parameters of the hot plastic deformation, the samples were cooled in water and air.

Hot-rolling of the tested steels sheets was performed on a single-frame reversing mill with a roll diameter of 550 mm. The linear speed of the rollers was a constant value of 0.74 m/s, while the actual plastic deformation rate for subsequent hot-rolling culverts was calculated according to Equation (1) given by Ekelunda [40].
(1)$$\dot{\varepsilon }=\frac{2V_{w}}{h_{1}+h_{2}}\cdot \sqrt{\frac{\Delta h}{R}},$$
where:

V_w_—equal peripheral speed of rollers 0.74 m/s;

h_1_, h_2_—thickness of before and after passes;

Δh—absolute crease;

R—roll radius, amounting to 0.275 m.

The plastic deformation rates for subsequent culverts were, respectively, 9.5, 10, 10.3, and 10.1 s^−1^. Similar plastic deformation rates were used for plastically deformed samples in the Gleeble simulator. It should be taken into account that under industrial conditions the plastic deformation rates in the final culverts reach values up to 50 s^−1^ and even up to 100 s^−1^. The parameters of the hot-rolling process were developed assuming that the size of the value of plastic deformation was determined by the allowable force of the band pressure on the roll. The basis for determining the pressure force during rolling was determining the average unit pressure. Proper description of this value for specific conditions always consists in choosing a method appropriate for a given process; therefore, after performing the appropriate analyses, it was considered that it was best to use the Zjuzin method to determine the average unit pressure exerted by the sheet on the rolling mill [41,42]. The advantage of this method is that it takes into account the state of stress and the width of the strand depending on the rolling coefficient’s shape factor. The coefficient of the influence of bandwidth was variable, depending on the ratio of the length of the contact arc projection to the average height of the band and the ratio of the average band width to its average height. The use of an appropriate function describing the plasticizing stress is of basic importance for the optimal design of the intensity of the cracks in the rolling process. The method developed by Hensel and Splitt was used to calculate the value of plasticizing stress [40,43]. The calculated parameters formed the basis for the development of the process control program, the basic parameters of which are presented in Table 2. The intervals between culverts ranged from 6 to 10 s and were selected so as to obtain the required hot deformation end temperature of 850 °C each time. Test sections of the tested steels with dimensions of 5 × 185 × 600 mm were subjected to hot-rolling. The batch was austenitized at 1150 °C for 15 min. Figure 6 shows the diagram of the actual hot-rolling process with three variants of final cooling.

The purpose of each cooling variant was:
Variant 1—supersaturation of steel after plastic deformation in the last culvert with a deformation value of 20%, under deformation strengthening conditions controlled by dynamic healing;Variant 2—cooling of steel in the air after plastic deformation in the last culvert with a deformation value of 20%, under deformation strengthening conditions controlled by static and metadynamic recrystallization;Variant 3—supersaturation of steel after plastic deformation in the last culvert with a 20% deformation value and isothermal annealing at its deformation temperature for 30 s, in conditions ensuring the assumed proportion of statically recrystallized austenite grains.

Samples for structural testing were prepared in two ways. Some samples for structural tests using an optical and a scanning electron microscope were taken, and then ground and mechanically polished. A 5% solution of HNO3 in ethyl alcohol was used as the reagent to reveal the structure. Samples for EBSD diffraction studies using a scanning electron microscope were mechanically ground and then electrochemically polished using a reagent with the following composition:
Total of 950 mL 99% acetic acid (CH_3_COOH);Total of 50 mL 60% perchloric acid (HClO_4_).

Observations of the structure of the tested steel were made on a Zeiss Axio Observer optical microscope (Jena, Germany), as well as on a SUPRA 35 scanning electron microscope (SEM, Jena, Germany) at an accelerating voltage of 20 kV using secondary electron detection (SE). In order to examine the chemical and phase composition of the samples’ micro-areas, as well as precipitates and particles, an EDS detector and a diffraction camera coupled to the abovementioned microscope in an Edrid Trident XM4 system were used. Tests using backscattered electron diffraction (EBSD) were performed at an accelerating voltage of 15 kV, a working distance of 17 mm, and a step size of 0.20 μm. 

Samples for structural examination using a transmission electron microscope (TEM) were cut and then prepared by mechanically grinding them to a thickness of 60 µm, followed by ion beam polishing. Transmission electron microscope studies were conducted at an accelerating voltage of 300 kV. TEM investigations were performed using a probe Cs-corrected FEI Titan 80-300 S/TEM microscope (Hillsboro, OR, USA) equipped with EDAX EDS. Selected area electron diffraction patters were obtained with a camera length of 215–330 mm and a C2-50 condenser aperture. In STEM mode, HAADF (high-angle annular dark field) was used with a convergence angle of 24 mrad.

The main purpose of the research was to analyze the structural changes of X98 and X105 steels after simulation of several hot compression stages and the actual hot-rolling of trial sheet sections, together with the determination of the structural processes occurring in the intervals between individual hot forming cycles. 

## 3. Results and Discussion

### 3.1. Plastometric Behavior

A detailed analysis of the σ–ε curves for the tested high-manganese TRIPLEX steels obtained on the Gleeble 3800 simulator indicated that the strain rate had a large impact on the value of the flow stress, which for the applied deformation conditions ranged from 110 to 485 MPa (Table 3, Figure 7). It was also observed that as the strain rate increased, the strain value increased at the maximum yield stress ε_max_, from 0.13 to 0.28. An effect of deformation temperature on ε_max_ was also noted (Table 3). For example, for X98 steel at a temperature of 1050 °C, the deformation ε_max_ was 0.18 and increased to about 0.25 with a decrease in the compression temperature to 850 °C (i.e., there was an increase in the value of plasticizing deformation by about 40%). However, in the case of X105 steel, the deformation ε_max_ had a value of 0.16 at 1150 °C and increased with the reduction in the compression temperature to 850 °C to about 0.28, so the plasticizing deformation values increased by about 75%. The initiation of dynamic recrystallization occurred before reaching the deformation ε_max_, corresponding to the maximum value of the plasticizing stress. Knowledge of these processes is important in order to precisely design the thermomechanical treatment process under industrial conditions. The initiation of dynamic recrystallization at 850 °C took place after a true strain of about 0.2 for both steels. The production of a fine-grained structure as a result of dynamic recrystallization, under industrial conditions, is therefore achievable provided that these thermomechanical treatment parameters are mapped onto industrial mills [32,33,34,35,36,37,44,45]. 

Detailed analysis of the stress–strain curve obtained in the four-stage hot compression of the tested TRIPLEX steels (Figure 8) allows us to state that the process controlling deformation strengthening in the first and second stages of hot plastic deformation at 1100 and 1050 °C, respectively, was dynamic recrystallization. While the analysis of the stress–strain curve in the first and second stages of plastic deformation did not clearly show this, when the results obtained in the one-stage plastic deformation at 1050 °C were taken into account, it was found that, for a deformation rate of 10 s^−1^, the value of the maximum plasticizing stress at which the dynamic recrystallization process was initiated was achieved at a deformation value of 0.183 for X98 steel and 0.163 for X105 steel. The plastic deformation applied for both the first (at 1100 °C) and the second (at 1050 °C) deformation stages was 0.23 (i.e., it was higher than the maximum yield stress determined in the one-stage compression of these samples). Additionally, the temperature of the first stage of deformation (1100 °C) was 50 °C higher than in the case of tests performed in the single-stage compression of the tested steels, while on the basis of these test results a clear trend could be seen that, along with the increase in deformation temperature at a constant rate of plastic deformation, in this case 10 s^−1^, the critical crush value decreased from 0.253 to 0.183 for X98 steel and from 0.279 to 0.163 for X105 steel. These changes were necessary to achieve maximum plasticizing stress and thus to initiate the process controlling the deformation strengthening, which was dynamic recrystallization. The change in the value of the plastic deformation rate from 7 s^−1^ in the first stage of deformation at 1100 °C, and its increase to a value of 10 s^−1^ in the last (fourth) stage of deformation, was an attempt to map the actual plastic deformation rates used on the rolling line. Based on the results of one-stage compression tests, it was found that the reduction in the deformation rate at a constant test temperature caused a decrease in the value of the deformation value. For example, for X98 steel, hot plastic deformation at 1050 °C, together with the reduction of the deformation rate from 10 to 1 s^−1^, the deformation value decreased from 0.183 to 0.166, this being necessary to achieve maximum plasticizing stress and thus to initiate the process of dynamic recrystallization that controlled the deformation strengthening. This trend was maintained for both the tested X98 and X105 steels in the range of the tested plastic deformation rate from 10 to 0.1 s^−1^. Based on this analysis, it was found that a plastic deformation rate of 7 s^−1^ in the first stage of deformation at 1100 °C and its increase to a value of 8 s^−1^ in the second stage of deformation were sufficient to initiate the process of dynamic recrystallization that controlled the deformation strengthening. In the next stage of deformation at 950 °C, the used value of 0.23 was on the border of the critical crush necessary to initiate dynamic recrystallization, which was determined in one-stage deformation tests of 0.233 and 0.214 (Table 3) for steels X98 and X105, respectively. However, it must be remembered that the process of dynamic recrystallization was already initiated in the first and second stages of the hot plastic deformation at temperatures of 1100 and 1050 °C, respectively, and reduction of the deformation temperature to 950 °C, as well as a slight increase in the deformation rate from 8 to 9 s^−1^, caused a continuation of the dynamic recrystallization process, which was also evidenced by the fact that the stress value during this deformation dropped to about 300 MPa, which is about 50MPa less than for the one-stage deformation at 950 °C (Figure 7 and Figure 8). In the last stage of plastic deformation, at a temperature of 850 °C, dynamic recrystallization was also a process controlling deformation strengthening, being a continuation of this process which was initiated in the earlier stages of plastic deformation, by analogy to that at a temperature of 950 °C. The analysis of σ–ε curves in the multistage hot compression allowed for the conclusion that in both the tested steels, between the first and second stage of plastic deformation, there was an increase in the value of the plasticizing stress from about 160 to 200 MPa for a deformation temperature of 1050 °C, and then to about 300 MPa for the third stage carried out at a temperature of 950 °C. The applied intervals between particular deformations allowed a partial course of static recrystallization. For the temperature of the last deformation, 850 °C, there was a more rapid increase in the plasticizing stress to a value of about 400–450 MPa, depending on the steel tested, which was associated with both a lower deformation temperature and a shorter break time between the third and fourth deformations of only 7 s. The limitation of the break time in simulated rolling resulted from the adaptation of the cooling conditions to the actual cooling time of ca. 2.5 mm thick sheets between the third and fourth deformation on production lines under real conditions. In addition, the increase in the stress values in the last (i.e., fourth) stage of deformation could have been influenced by the fact that the material was already partially fragmented as a result of the dynamic recrystallization in the first and second stages of deformation. The applied conditions of the multistage deformation resulted in fragmentation occurring, mainly of austenite grains, and a change in the ferrite morphology in the tested TRIPLEX steels [32,33,34,35,36,37,44], which had already been noted in previous publications [46,47].

### 3.2. Microstructure 

Figure 9 shows steel structures X98 (Figure 9a) and X105 (Figure 9b) after free forging. Based on observations of the structure using light and electron microscopy (SEM), it was found that the examined steels were characterized by an austenitic-ferritic structure with carbides, as detailed in other publications [17,18]. Based on the observation of the structure of the analyzed newly developed steels after forging, it was found that in X98 steel ferrite grains were evenly distributed at the austenite grain boundaries. In X105 steel, it was noted that the ferritic areas were definitely larger (longer and wider) than in X98 steel, and their banding arrangement was also noted. Analyzing the above research results, it can be stated that the Nb and Ti additions affect austenite grain refinement. The elements Nb and Ti forming dispersion nitrides, carbonitrides, and carbides, under appropriately selected conditions, were the reason for the additional solidification of the steel. In addition, in hot deformed austenite, they inhibited the growth of recrystallized austenite grains, which contributed to the formation of a fine-grained structure. Research carried out using the EBSD technique (Figure 10 and Figure 11) allowed the share, distribution, and location of individual structural components to be illustrated, along with their morphology and size. The average grain diameter of austenite in the forged state of X98 steel was 42 µm, while in X105 steel it was 62 µm (Figure 12a). Based on the EBSD study, it can be concluded that both steels have wide-angle boundaries with a misorientation angle Ѳ > 15°, whose average percentage is similar and amounts to ~94% (Figure 10c, Figure 11c, and Figure 12b).

The use of a four-stage hot compression of axisymmetric samples in the Gleeble simulator, with a true strain equal to 4 × 0.23 for X98 and X105 TRIPLEX steels, caused (after the last deformation at 850 °C) the main processes controlling deformation strengthening to be dynamic recrystallization and static recrystallization, occurring between subsequent deformations (Figure 8, Figure 13 and Figure 14). The average diameter of dynamically and statically recrystallized austenite grains for thermomechanical treated, water-cooled X98 steel after the last deformation at 850 °C was 10 μm for X98 steel and 16 μm for X105 steel (Figure 13a and Figure 14a). As a result of the applied thermomechanical treatment, fine grains of ferrite in the X98 steel occurring at the austenite grain boundaries in the forged state (Figure 9) merged to form new larger grains, while in X105 steel ferrite grains elongated in a direction perpendicular to the compression direction (Figure 14). The tested steels subjected to free cooling in the air after the last deformation at 850 °C had an average austenite grain diameter of 14 μm for X98 steel and 21 μm for X105 steel, which indicated a greater proportion of metadynamic recrystallization resulting in an increase in grain size (Figure 13b and Figure 14b). Isothermal heating of the steel after the last plastic deformation at a temperature of 850 °C for 30 s in accordance with variant 3 of the thermomechanical treatment caused the main processes removing the effects of deformation strengthening to be metadynamic and static recrystallization, as a result of which the average diameters of the austenite grains were 8 μm for X98 steel and 13 μm for X105 steel (Figure 13c and Figure 14c). The reduction in ferrite grain size is clearly visible here compared to variants 1 and 2. Figure 15 (X98 steel) and 16 (X105 steel) show the results of the EBSD technique. The crystallographic orientation maps use the IPF (inverse pole figures) color scheme, thanks to which it is possible to show the orientation of crystallographic directions in individual grains in relation to the coordinate system adopted for the examined sample. Figure 15a and Figure 16a show the orientation maps of both tested steels after hot compression with the compression direction indicated. In both steels there were two characteristic areas associated with the occurrence of two basic ferrite orientations [111] and [101]. Twins [001] and [101] with respect to parent grains in X98 steel were disclosed. The structure of X98 steel was characterized by a greater grain refinement compared to X105 steel, which is caused similarly, as already indicated above, in the initial state (after forging) by the addition of Nb and Ti, which form dispersive particles of nitrides, carbonitrides, and carbides with a regular network, limiting the grain growth of the recrystallized austenite. Mn_7_C_3_ carbide precipitations at grain boundaries were also identified in both the steels examined. The results of the EBSD technique for X98 and X105 steel confirmed the dominance of wide-angle boundaries in the structure: In X98 steel, their average percentage exceeded 80%; and in X105 steel, it reached a value close to 65% (Figure 12b, Figure 15b, and Figure 16b). In both tested steels there were twins of deformation with a disorientation angle of 58°–62°: In X98 steel their share was ~18%, while in X105 steel their share was smaller and amounted to ~11%. The relatively large share of low-angle boundaries (misorientation angle Ѳ < 15°) in X98 and X105 steels, which were ~20% and ~35%, respectively, may indicate that recrystallization in the studied steels had not come to the end (Figure 12b, Figure 15b, and Figure 16b).

The structures of X98 and X105 TRIPLEX steels after four-stage hot-rolling with a true strain equal to 0.23 for each culvert, according to the parameters presented in Figure 6, are shown in Figure 17 and Figure 18. Structural analysis was carried out on samples cut in accordance with the rolling direction. The average diameter of austenite grain after hot-rolling and cooling according to variant 1 in X98 steel was 27 µm, while in X105 steel it was 38 µm. The structures of both steels were austenite grains with numerous annealing and deformation twins. Additionally observed were ferrite bands in the form of elongated grains in both steels, which is a characteristic feature of austenitic-ferritic structures, resulting from a low tendency to recrystallize ferrite. These bands were definitely longer, wider, and more visible in X105 steel. In X98 steel, the ferrite bands were much smaller and more strongly defragmented. The aluminum content, ~11% in the examined steels, promoted the formation of ferrite strands parallel to the rolling direction [2]. Based on the results of EBSD, it was found that the structures of the samples of both steels after hot-rolling were characterized by two main orientations in the direction [111] and [001], and the proportion of grains oriented in the direction [111] parallel to the hot-rolling direction (HRD) was about 70% (Figure 19a and Figure 20a). The presence of Mn_7_C_3_ carbide at grain boundaries in both steels was also noted. Based on EBSD research, it was found that the four-stage heat and plastic deformation treatment completed with cooling variants 1–3 in both steels was characterized by obtaining a structure with the dominance of wide-angle borders, whose percentage was generally smaller than after deformation using the Gleeble simulator and ranged from 60% to 71% for X98 steel and 47% to 71% for X105 steel (Figure 12b, Figure 19b, and Figure 20b). In steels after hot-rolling, the share of deformation twins compared to steels after four-stage hot compression in the Gleeble 3800 simulator was smaller and amounted to 7% to 16% for X98 steel and 5% to 8% for X105 steel (Figure 12). The share of low-angle boundaries (misorientation angle Ѳ ≤ 15°) in X98 steel was 29%–40%, and their share was similar to X105 steel (29%–53%) in cooling variants 1 and 3 (Figure 12). The share of low-angle boundaries in the second cooling variant, with a particularly high share of non-recrystallized grains in the X105 steel, was clearly differentiated in favor of X98 steel. The large share of low-angle boundaries with a misorientation angle of less than 15° determined whether the recrystallization in the steels had come to the end. X98 steel, after hot-rolling and cooling according to variants presented in Figure 6, was characterized by a greater grain fragmentation than X105 steel, similarly to the case of uniaxial four-stage hot compression in the Gleeble thermomechanical simulator (Figure 12 and Figure 21). 

Tests using the SEM equipped with an X-ray energy dispersion spectrometer allowed the identification of dispersion carbides based on Nb and Ti in X98 steel, which were released in austenite, ferrite, and the grain boundaries (Figure 22, Table 4). The size of the identified carbides based on Nb and Ti ranged from a few nanometers to 15 µm [18]. At the grain boundaries in both steels, AlN precipitations of 1 to 3 µm were also noted (Figure 23, Table 5). The niobium and titanium in X98 steel were bound in carbides and effectively inhibited the growth of austenite grains. At the boundaries of austenite and ferrite grains in both the examined steels, the occurrence of κ-(Fe, Mn)_3_AlC carbides was also noted (Figure 22a and Figure 23). In X98 steel, the size of the κ-carbides ranged from several nanometers to about 160 nm, while in X105 steel both carbides with a size of several nanometers and those up to 800 nm appeared. In steels cooled according to variant 2 (air-cooling), the average κ-carbide size was four times greater than that of water-cooled steels. These carbides can be the cause of steel brittleness during plastic deformation at room temperature when they form large precipitates [2,26,48,49,50,51,52] at grain boundaries. TEM tests allowed the identification of κ-carbide in X98 steel (Figure 24) with a regularly face-centered cubic (fcc) network (Pm-3m group) and a network parameter a = 0.3875 nm [28]. In X105 steel, Mn_7_C_3_ carbide was identified in austenite (Figure 25), characterized by an orthorhombic crystal lattice (Pnma group) with lattice parameters a = 0.4546 nm, b = 0.6959 nm, and c = 1.197 nm. Mn_7_C_3_ carbide has also been identified in X98 steel, as described in earlier publications [17,18,21,41,43,46,47]. Mn_7_C_3_ carbides occur in both austenite and ferrite, and their size ranges from 100 to 600 nm.

## 4. Conclusions

Based on the results of our research and analysis, the following conclusions can be drawn:
The structure of the newly developed TRIPLEX steels after forging consisted of austenite, ferrite, and carbides. The average grain diameter of austenite in the forged state of X98 steel was 42 µm, while in X105 steel it was nearly 50% larger and amounted to 62 µm. The ferrite share in steel with Nb and Ti (X98) additions was on average around 11%, and in the case of the reference X105 steel its share was definitely higher and amounted to about 27%;After hot plastic deformation using the Gleeble simulator and a semi-industrial rolling line, the structure of both tested steels was similar in terms of phase composition. X98 steel, due to its Nb and Ti content, was characterized by a significantly smaller size of austenite grain and a share in the structure of, among others compound carbides, (Nb, Ti)C. However, both the tested steels differed significantly, as in the initial state after forging, with regard to the participation and arrangement of ferrite. In X98 steel, after simulated plastic deformation (Gleeble) with different cooling variants, ferrite was quite evenly distributed in its structure, with the largest areas of ferrite revealed in option 2. In X105 steel, ferrite occurred in the form of elongated areas in the direction of rolling in variants 1 and 2, and fine grains in option 3. These clear differences in the distribution and form of ferrite in the simulated deformations no longer found their analogy in the structure of the steel after rolling, where the ferrite in all cooling variants was arranged in the form of highly-elongated strands in the direction of rolling, which resulted from a low tendency to undergo recrystallization. In addition, the formation of ferrite bands parallel to the rolling direction was affected by the high concentration of aluminum in the steels tested, which was also noted by Bausch et al. [2] during their research. The ferrite bands in X105 steel were definitely wider, which was related to the grain size in the analyzed steels, while the grain in X98 steel was smaller in all machined variants;It was found that the process controlling deformation strengthening at all stages of the hot plastic deformation was dynamic recrystallization, together with static recrystallization in the intervals between subsequent stages, especially between the last stages. Cooling of the analyzed steels after thermomechanical treatment in air promoted metadynamic recrystallization and an increase in the average grain size. The applied isothermal annealing after plastic formation caused fragmentation of the structure, because the main processes removing the effects of deformation strengthening were metadynamic and static recrystallization;On the basis of EBSD tests in each of the presented states (after forging, after hot compression, and after hot-rolling), it was found that the studied steels were dominated by wide-angle boundaries (misorientation angle Ѳ ≥ 15°), with the exception of X105 steel from the second cooling variant after rolling. Deformation twins with a misorientation angle of 58°–62° were also disclosed, while for most variants with X98 steel their share was definitely greater. The share of low-angle boundaries of about 20%–30% (misorientation angle Ѳ < 15°) in both the tested steels may indicate that the recrystallization process was not completed with the adopted thermomechanical treatment plan;Research on the structure of the X98 and X105 steels using a transmission electron microscope allowed the identification of M_7_C_3_-type carbides and nanometric κ-(Fe, Mn)_3_AlC carbides located inside the austenite and ferrite grains, as well as at the grain boundaries. Only in variant 2, cooling after heat treatment, were the carbides in question definitely larger and located mainly on grain boundaries, which may significantly reduce the mechanical properties of the steels tested after this type of treatment. M_7_C_3_-type carbides with orthorhombic crystal lattices were revealed in both the steels examined in the austenite and ferrite grains;It was also found, on the basis of tests using SEM coupled with an EDS spectrometer, that the X98 steel had dispersive carbides based on Nb and Ti, which were released in austenite, ferrite, and the grain boundaries. The size of these carbides ranged from several nanometers to about 15 µm. At the grain boundaries in both the analyzed steels, AlN precipitations of up to 3 µm were also noted. At the boundaries of austenite and ferrite grains in both the investigated steels, κ-(Fe, Mn)_3_AlC carbides were also found. In X98 steel, the size of the κ-carbide was smaller and ranged from a few to 160 nm, while in X105 steel both carbides had a size ranging from several nanometers up to nearly 1 µm.

## Figures and Tables

**Figure 1 materials-13-00739-f001:**
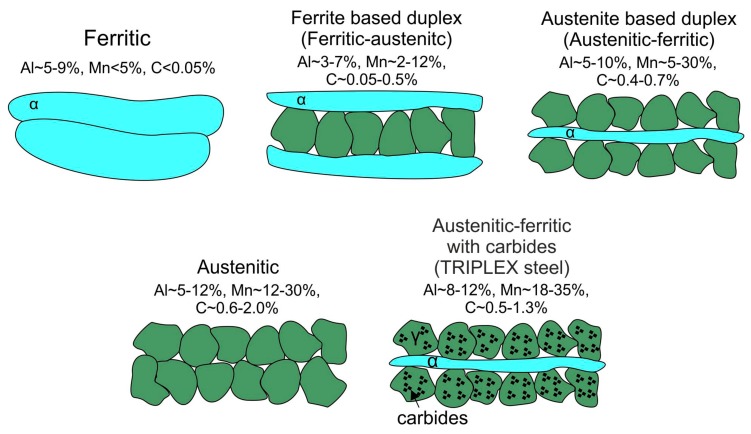
Division into different structures of steel with reduced density after hot-rolling (developed on the basis of [1,2,4]).

**Figure 2 materials-13-00739-f002:**
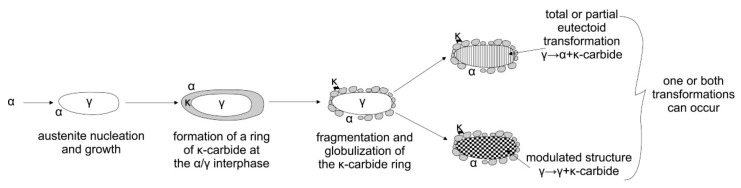
Schematic representation of phase transformation in Fe-Mn-Al-C steels [25,26].

**Figure 3 materials-13-00739-f003:**
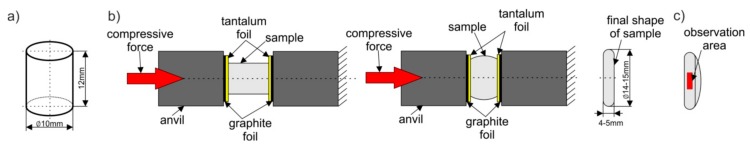
Scheme of the sample for plastometric tests with dimensions (**a**), scheme of the compression test with the final shape of the sample (**b**), marked area of the specimen subjected to the detailed structural studies (**c**).

**Figure 4 materials-13-00739-f004:**
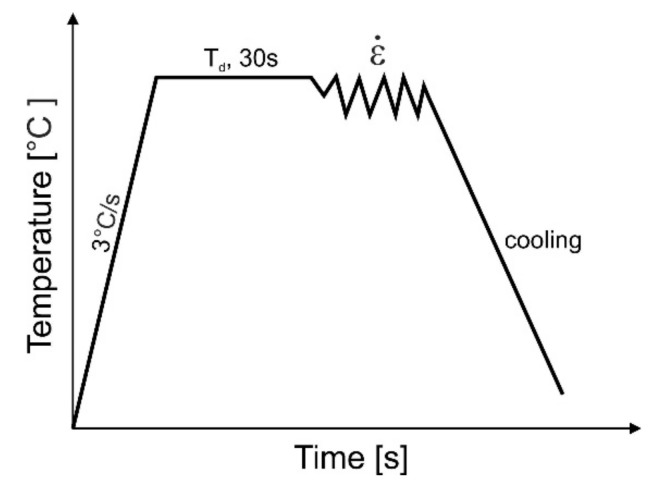
Scheme of the compression process of axisymmetric samples, T_d_ = 850, 950, 1050 °C; ε = 1; $\dot{\varepsilon }$ = 0.1, 1, 10 s^−1^

**Figure 5 materials-13-00739-f005:**
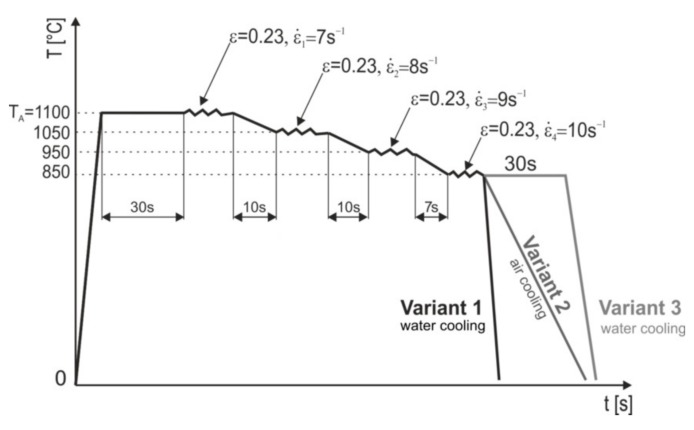
Scheme of four-stage thermomechanical treatment for axisymmetric samples of tested plastically deformed steels in the Gleeble 3800 simulator.

**Figure 6 materials-13-00739-f006:**
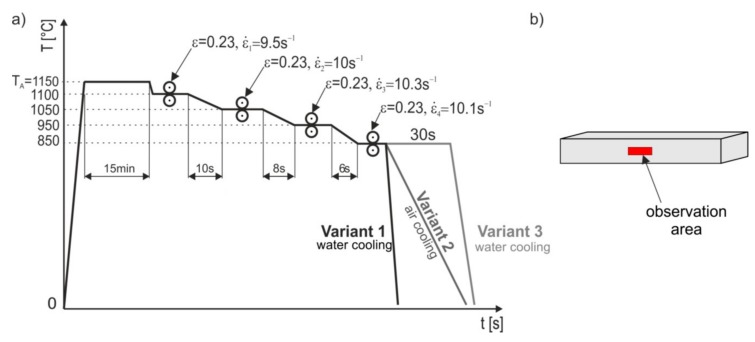
Variants of hot plastic treatment with different cooling of the actual hot-rolling process (**a**), marked area of the hot-rolled specimen subjected to the detailed structural studies (**b**).

**Figure 7 materials-13-00739-f007:**
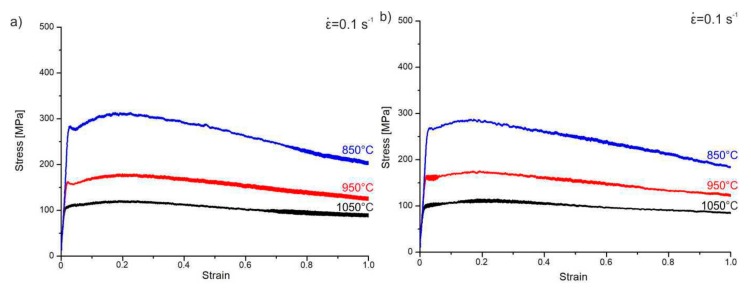
The influence of strain rate and temperature on the shape of the stress–strain curves of tested steels after true strain: ε = 1, the rate of plastic deformation: (**a**,**b**) $\dot{\varepsilon }$ = 0.1 s^–1^, (**c**,**d**) $\dot{\varepsilon }$ = 1 s^–1^, (**e**,**f**) $\dot{\varepsilon }$ = 10 s^–1^; (**a**,**c**,**e**) for X98 steel, (**b**,**d**,**f**) for X105 steel [39].

**Figure 8 materials-13-00739-f008:**
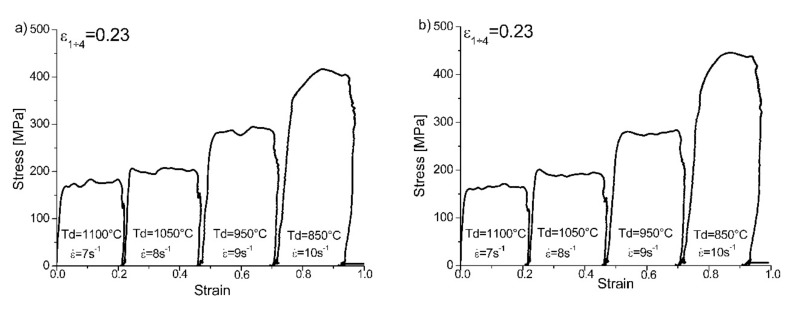
Stress–strain curves of four-step hot compression with true strain 4 × 0.23 axisymmetric samples from X98 (**a**) and X105 (**b**) steels, according to the thermomechanical treatment scheme shown in Figure 4.

**Figure 9 materials-13-00739-f009:**
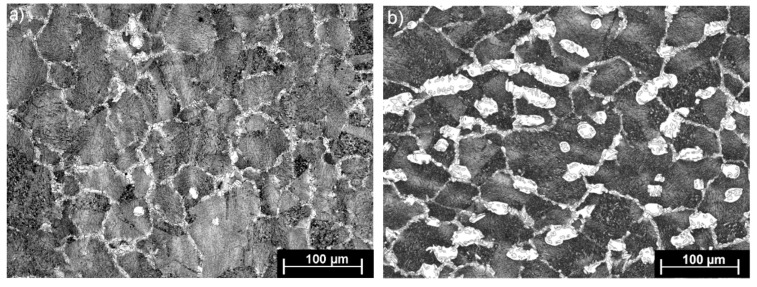
Austenitic ferritic structure of X89 (**a**) and X105 (**b**) steels after forging.

**Figure 10 materials-13-00739-f010:**
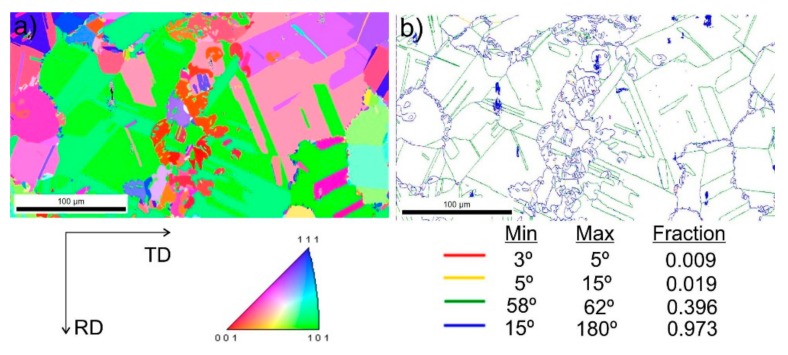
Structure revealed by EBSD technique in SEM in the selected micro-area of the investigated X98 steel after forging: Crystallographic orientation map (**a**), misorientation angles map (**b**).

**Figure 11 materials-13-00739-f011:**
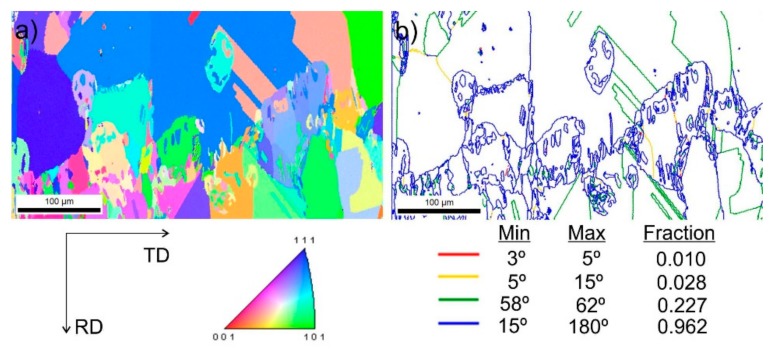
Structure revealed by EBSD technique in SEM in the selected micro-area of the investigated X105 steel after forging: Crystallographic orientation map (**a**), misorientation angles map (**b**).

**Figure 12 materials-13-00739-f012:**
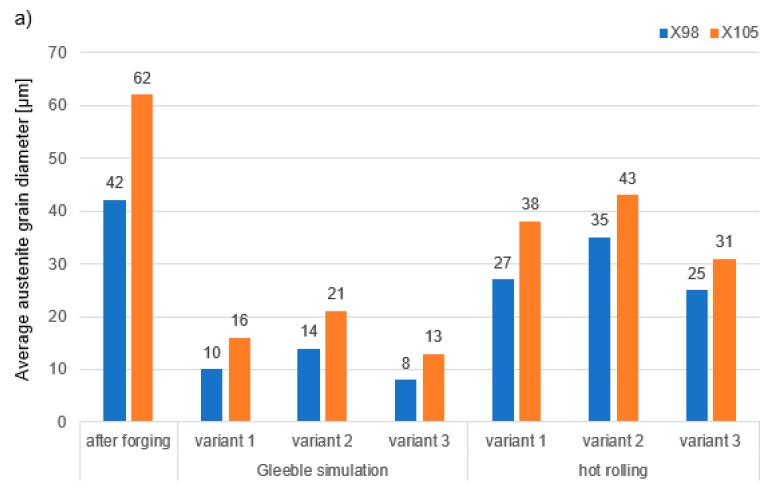
Average grain diameter of austenite (**a**) and misorientation angles (**b**) in the analyzed X98 and X105 steels: After forging, Gleeble simulation, and hot-rolling.

**Figure 13 materials-13-00739-f013:**
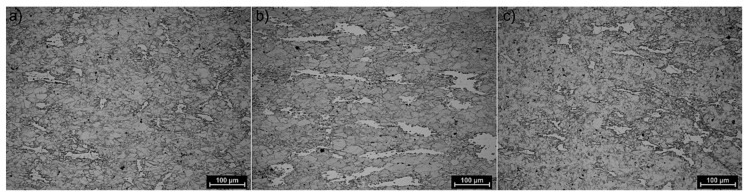
X98 steel structures after four-step hot compression in the Gleeble 3800 simulator and cooling according to variant 1 (**a**), variant 2 (**b**), and variant 3 (**c**).

**Figure 14 materials-13-00739-f014:**
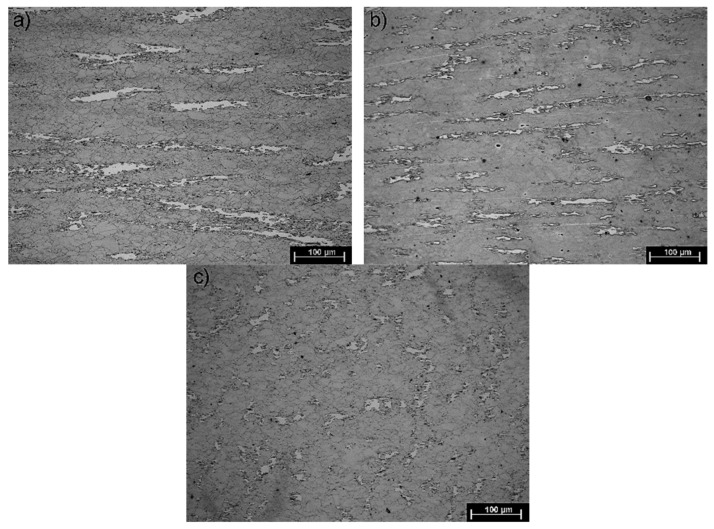
X105 steel structures after four-step hot compression in the Gleeble 3800 simulator and cooling according to variant 1 (**a**), variant 2 (**b**), and variant 3 (**c**).

**Figure 15 materials-13-00739-f015:**
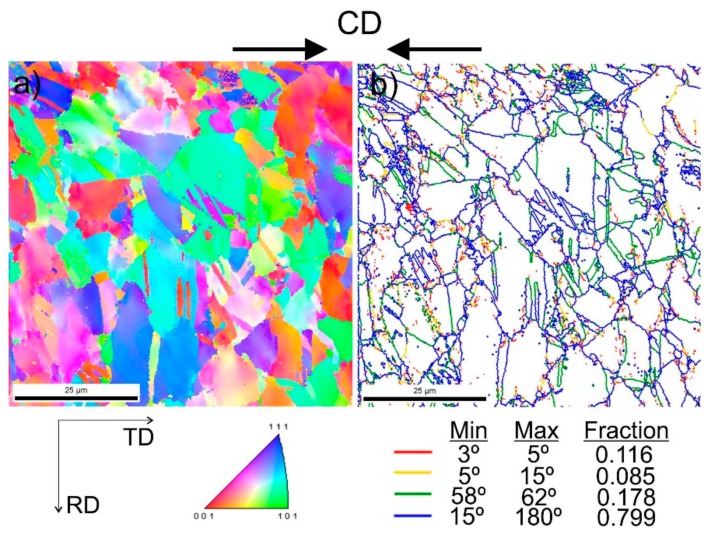
Structure revealed by EBSD technique in SEM in the selected micro-area of the investigated X98 steel after Gleeble simulation and cooling in water (variant 1): Crystallographic orientation map (**a**), misorientation angles map (**b**); CD—compression direction.

**Figure 16 materials-13-00739-f016:**
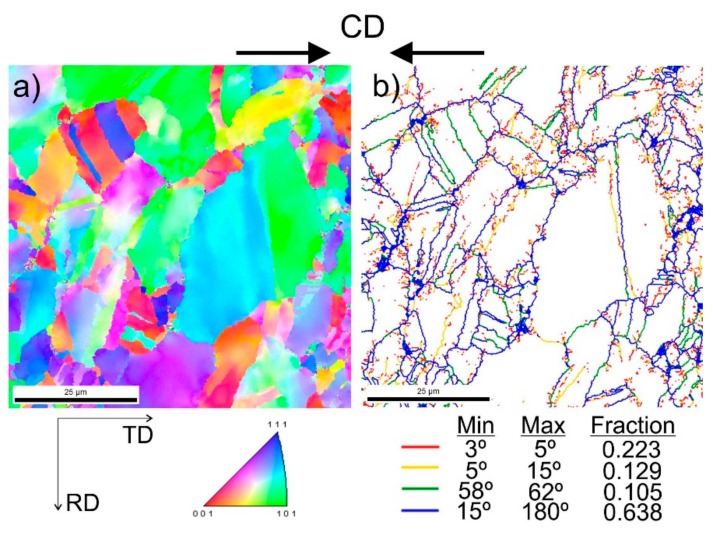
Structure revealed by EBSD technique in SEM in the selected micro-area of the investigated X105 steel after Gleeble simulation and cooling in water (variant 1): Crystallographic orientation map (**a**), misorientation angles map (**b**); CD—compression direction.

**Figure 17 materials-13-00739-f017:**
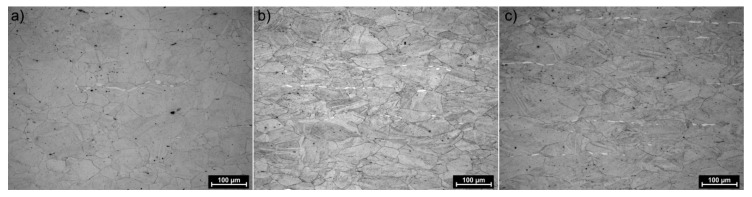
X98 steel structure after four-step hot-rolling and cooling according to variant 1 (**a**), variant 2 (**b**), and variant 3 (**c**).

**Figure 18 materials-13-00739-f018:**

X105 steel structure after four-step hot-rolling and cooling according to variant 1 (**a**), variant 2 (**b**), and variant 3 (**c**).

**Figure 19 materials-13-00739-f019:**
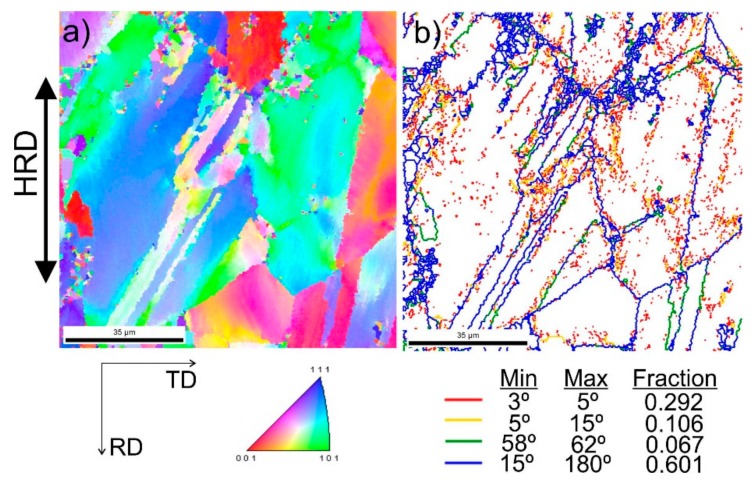
Structure revealed by EBSD technique in SEM in the selected micro-area of the investigated steel X98 after hot-rolling and cooling in water (variant 1): Crystallographic orientation map (**a**), misorientation angles map (**b**); HRD—hot-rolling direction.

**Figure 20 materials-13-00739-f020:**
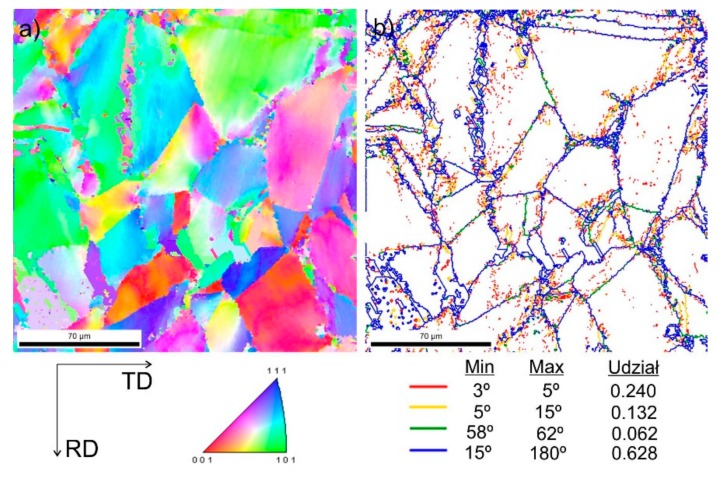
Structure revealed by EBSD technique in SEM in the selected micro-area of the investigated steel X105 after hot-rolling and cooling in water (variant 1): Crystallographic orientation map (**a**), misorientation angles map (**b**).

**Figure 21 materials-13-00739-f021:**
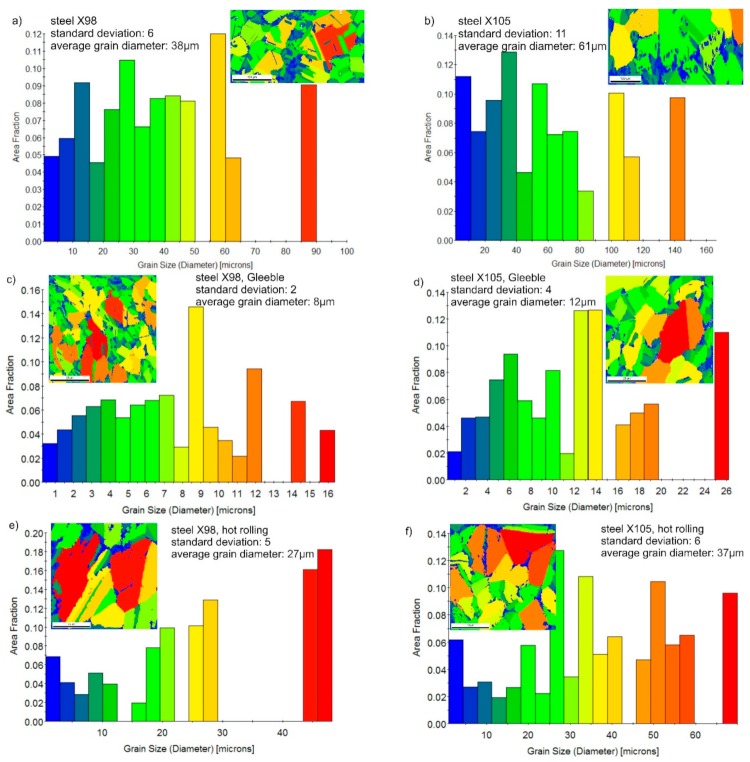
Grains size revealed by EBSD technique in SEM in the selected micro-area of the investigated X98 (**a**,**c**,**e**) and X105 (**b**,**d**,**f**) steels: After forging (**a**,**b**), after Gleeble simulation and cooling in water (variant 1) (**c**,**d**), and after hot-rolling and cooling in water (variant 1) (**e**,**f**).

**Figure 22 materials-13-00739-f022:**
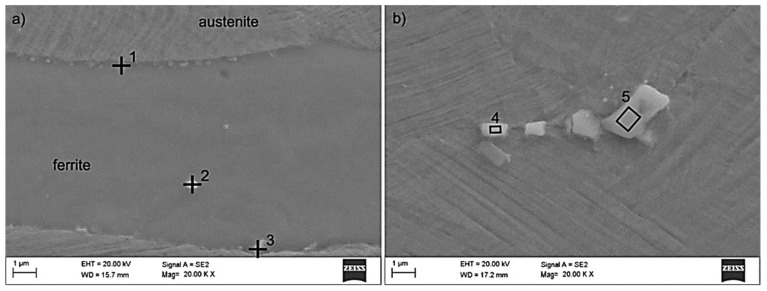
X98 steel structure after hot-rolling and cooling: According to variant 3 (**a**), according to variant 5 (**b**) with the revealed κ-carbides and (Nb, Ti) C together with the measurement in the marked micro-areas of the chemical composition by the EDS technique (SEM).

**Figure 23 materials-13-00739-f023:**
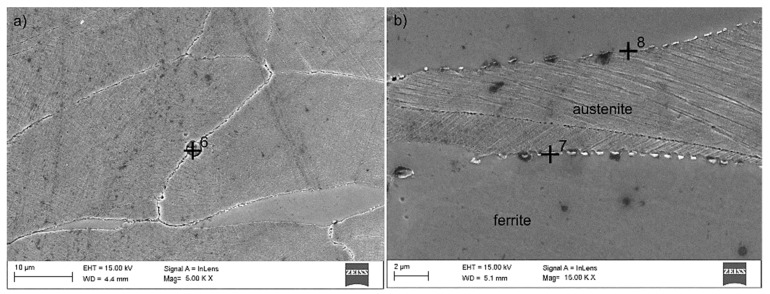
X105 steel structure after hot-rolling and cooling: According to variant 2 (**a**), according to variant 6 (**b**) with the revealed κ- carbides and AlN together with the measurement in the marked micro-areas of the chemical composition by the EDS technique (SEM).

**Figure 24 materials-13-00739-f024:**
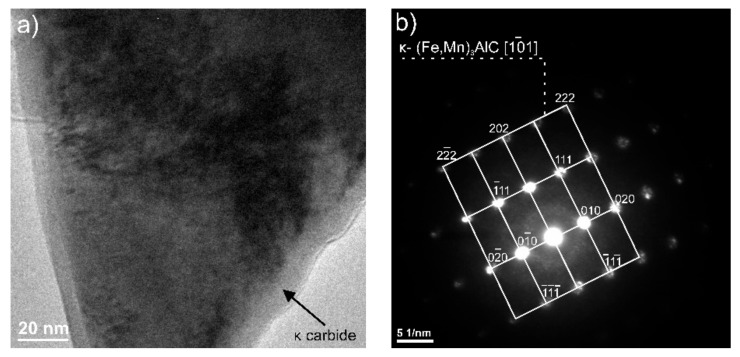
TEM image of the X98 steel κ-carbide precipitates (**a**), diffraction pattern of the zone axis [101] κ (**b**).

**Figure 25 materials-13-00739-f025:**
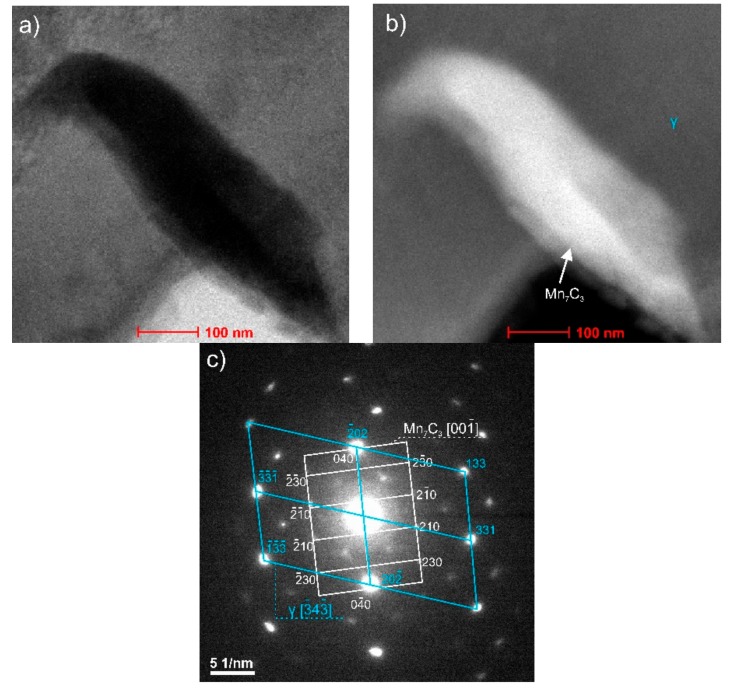
Precipitation of Mn_7_C_3_ carbide in austenite – X105 steel: Dark field image (**a**), bright field image (**b**), diffraction pattern of the zone axis for [343] austenite and [001] Mn_7_C_3_ (**c**).

**Table 1 materials-13-00739-t001:** Chemical composition of the investigated steel (wt.%).

C	Mn	Al	Si	Nb	Ti	Ce	La	Nd	P_max_	S_max_
**X98MnAlNbTi24-11 steel (X98 steel)**										
0.98	23.83	10.76	0.20	0.048	0.019	0.029	0.006	0.018	0.002	0.002
**X105MnAlSi24-11 steel (X105 steel)**										
1.05	23.83	10.76	0.10	-	-	0.037	0.011	0.015	0.005	0.005

**Table 2 materials-13-00739-t002:** Rolling program for steel test sections.

Culvert No	The Temperature of Plastic Deformation T_d_ [°C]	Thickness before Culvert L_0_ [mm]	Thickness after Culvert L_1_ [mm]	The Absolute Degree of Crushing L_0_ − L_1_ [mm]	True Strain lnL1L0
**1**	1100	5	4	1.0	
**2**	1050	4	3.2	0.8	0.23
**3**	950	3.2	2.55	0.65	0.23
**4**	850	2.55	2	0.55	0.23

**Table 3 materials-13-00739-t003:** The influence of temperature and strain rate on the flow stress of tested high manganese TRIPLEX steels [41].

Strain Temperature	$\dot{\varepsilon }$ [s^−1^]	ε_max_		σ [MPa]	
		**Steel X98**	**Steel X105**	**Steel X98**	**Steel X105**
**850 °C**	0.1	0.178	0.127	310	284
	1	0.191	0.168	384	377
	10	0.253	0.279	485	476
**950 °C**	0.1	0.169	0.159	175	173
	1	0.195	0.163	257	246
	10	0.233	0.214	363	353
**1050 °C**	0.1	0.166	0.152	118	110
	1	0.169	0.156	170	159
	10	0.183	0.163	254	238

**Table 4 materials-13-00739-t004:** Results of the EDS spectrum analysis for the areas from Figure 22 (wt.%).

Element	Point 1	Point 2	Point 3	Area 4	Area 5
**C ^1^**	7	12	7	17	16
**Al**	14	10	14	1	4
**Nb**	-	16	-	48	32
**Ti**	-	6	-	17	12
**Mn**	22	15	22	9	13
**Fe**	57	41	57	8	23
**precipitation**	κ	(Nb,Ti)C	κ	(Nb,Ti)C	(Nb,Ti)C

^1^ C content is an approximate value due to the method of measurement.

**Table 5 materials-13-00739-t005:** Results of the EDS spectrum analysis for the areas from Figure 23 (wt.%).

Element	Point 6	Point 7	Point 8
**C ^1^**	12	8	6
**N ^1^**	8	-	-
**Al**	29	13	13
**P**	2	-	-
**S**	2	-	-
**Mn**	15	23	25
**Fe**	32	56	56
**precipitation**	AlN	κ	κ

^1^ C and N content is an approximate value due to the method of measurement.

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
