# Peer review of "Structure of Fe-Mn-Al-C Steels after Gleeble Simulations and Hot-Rolling"

_materials, 2020, doi:10.3390/ma13030739_

Round 1
Reviewer 1 Report
The manuscript discusses the structural changes of the newly developed TRIPLEX steels, Fe-Mn-Al-C(X105) and Fe-Mn-Al-NB-Ti-C(X98) after hot-rolling and thermomechanical treatment in Gleeble simulator. The manuscript is well-constructed, and the data and finding were presented clearly. The experimental methods were described in detail, and the results is clearly presented. To the best of my knowledge, this is the first study to have performed such detailed thermomechanical treatment and rolling on the two alloys of interest. I think this work is well-executed, especially the grain/structural analysis, and the supporting reasoning is very logical. I’m only providing minor comments that I believe will help improve the reader’s understanding.
Minor comments:
Line 121, Td can be defined here. … 3°C/s to a plastic 120 deformation temperature, Td, of 850, 950, …
In table 2, the “real deformation” is mention for the first time, the author should clarify in the main text how it is measured.
For Table 3, an alternative to a table, a double Y-axis scatter plot might visualize the finding more clearly than the table, not necessarily better – just an idea.
Author Response
Dear Reviewer,
Thank You very much for this review and all comments, which are very important for us and allow us to improve the quality of this publication and help us not to make such mistakes in the future.
Minor comments:
Line 121, Td can be defined here. … 3°C/s to a plastic 120 deformation temperature, Td, of 850, 950, …
Thank you for this remark, this has been corrected in the article text
In table 2, the “real deformation” is mention for the first time, the author should clarify in the main text how it is measured.
Thank you for this remark, the term “real deformation” was not correctly used by me at this ariticle which is focused on plastic deformation, therefore in table 2 as well as in the text this term “real deformation” was corrected to “true strain”. Of course true strain is determined as ln(l1/l0) what was also described in that table. Now I think it will be clear for readers.
For Table 3, an alternative to a table, a double Y-axis scatter plot might visualize the finding more clearly than the table, not necessarily better – just an idea.
Thank you for this comment, maybe in the next publication I will try to show this or similar results in the form of a chart.

Reviewer 2 Report
This is a very interesting paper reporting the microstructure formation in a Fe-Mn-Al-C steels via the actual rolling process and the compression tests by Gleeble simulator. The relationship between process parameters (deformation temperature, strain rate, strain, cooling rate, multi-compression) and microstructure has been studied through many experiments, and the microstructures are analyzed in detail. However, the explanation about experiments and the information on the nonuniform distribution inside the sample after compression test and rolling are insufficient.
Chapter 2:Show transformation point Ar3 for both samples.
In Figure 3:Graphite and tantalum foils are used between the sample surface and the anvil one. Why were the different foils used?
In Figure 4:Show prior austenite grain size at 850, 950 and 1050°C. Austenite grain size before deformation is one of important factors on microstructure formation.
The reviewer think that the strain has a distribution of a sample after a plastic deformation. Even if compression ratio is the same, the strain imposed in a sample is different by anvil compression test and rolling process. How do you think about it?
About microstructure : Show observation site in the sample after compression test and rolling.
In Figure 6 : How were a temperature measured and controlled in the rolling process?
In Figure 7 : Which is the s-s curve nominal or true?
Author Response
Dear Reviewer,
Thank You very much for this review and all comments, which are very important for us and allow us to improve the quality of this publication and help us not to make such mistakes in the future.
Chapter 2:Show transformation point Ar3 for both samples.
Thank you for this remark/comment. For this group of steels characterized by a high content of Mn and Al it is impossible to determine the transformation point Ar3, because the element Al is a ferrite-forming element and even at a temperature close to the melting point of about 1300°C the percentage of ferrite is about 34%, at temperature 1100 percentage of ferrite is about 26%. At room temperature depend on the type of steel is between 15 to 24%.
In Figure 3:Graphite and tantalum foils are used between the sample surface and the anvil one. Why were the different foils used?
Thank you for this remark/comment. I have wrongly marked the foils (colors and arrows) in the drawing; it was corrected on the figure 3. We always install graphite and tantalum foils then samples then tantalum and graphite foils again. We use graphite films to reduce friction, while tantalum films to protect against diffusion and welding with tungsten carbide anvils.
In Figure 4:Show prior austenite grain size at 850, 950 and 1050°C. Austenite grain size before deformation is one of important factors on microstructure formation.
The reviewer think that the strain has a distribution of a sample after a plastic deformation. Even if compression ratio is the same, the strain imposed in a sample is different by anvil compression test and rolling process. How do you think about it?
Thank you for this remark/comment. Of course, I agree with the reviewer's opinion. In ours earlier scientific publications (11, 20-21, 43-45) the influence of temperature, strain rate and plastic deformation as well as cooling after deformation have been described in details. Also some simulation of hot rolling in plain strain condition using also Gleeble simulator was used and obtained results have already been published. Due to the limited number of pages, we could not show all the results of the research, only we refer to these results in the discussion of the results
About microstructure : Show observation site in the sample after compression test and rolling.
Observation area have been marked on the figure 3 for specimen tested on the Gleeble (Fig. 3c), and on Fig. 6b was added extra drawings with marked area which was subjected for detailed structural studies.
In Figure 6 : How were a temperature measured and controlled in the rolling process?
The temperature was measured by pyrometers installed before entry and exit from the rolling mill and on the basis of the mean value and several preliminary tests was determined at a given value level
In Figure 7 : Which is the s-s curve nominal or true?
Thank you for this remark/comment. This is a true strain-true stress curves, the description of the results in the text has been corrected
